# Primary Thyroid Lymphoma: How Molecular Biology and Ancillary Techniques Can Help the Cytopathologist Overcome This Diagnostic Challenge

**DOI:** 10.3390/jpm13081203

**Published:** 2023-07-28

**Authors:** Leo Guidobaldi, Concetta Cafiero, Gerardo D’Amato, Marco Dell’Aquila, Pierpaolo Trimboli, Raffaele Palmirotta, Salvatore Pisconti

**Affiliations:** 1Cytodiagnostic Unit, Section of Pathology Sandro Pertini Hospital, ASL RM2, 00157 Rome, RM, Italy; leoguidobaldi@gmail.com; 2Medical Oncology, SG Moscati Hospital, 74010 Statte, TA, Italy; concettacafiero@gmail.com (C.C.); salvatorepisconti@hotmail.it (S.P.); 3Dipartimento di Scienze Biomediche Avanzate, Università degli Studi di Napoli Federico II, 80131 Naples, NA, Italy; 4Pathology Unit, Belcolle Hospital, ASL Viterbo, 01100 Viterbo, VT, Italy; mzrk07@gmail.com; 5Servizio di Endocrinologia e Diabetologia, Ente Ospedaliero Cantonale (EOC), 6500 Bellinzona, Switzerland; pierpaolo.trimboli@eoc.ch; 6Interdisciplinary Department of Medicine, School of Medicine, University of Bari “Aldo Moro”, 70124 Bari, BA, Italy

**Keywords:** thyroid, malignant lymphoma, fine-needle aspiration, cytology

## Abstract

Primary thyroid lymphoma (PTL) occurs rarely, its diagnosis is a challenge, and the prognosis of these patients depends on the time of diagnosis. Even though fine-needle aspiration cytology (FNAC) is recognized as the most accurate tool for detecting thyroid malignancies, its sensitivity for PTL is poor. Both clinical and ultrasound presentation of PTL can be atypical, and laboratory tests fail to furnish relevant data. Consequently, the reliability of a cytopathologist facing PTL can be poor, even when he is aware of its clinical information. In addition, the cases described in the literature are extremely rare and fragmentary, and consequently, the molecular data currently available for this neoplasm are practically negligible. Here, we present a case report in order to discuss the intrinsic limitations in achieving a final diagnosis of PTL and how using molecular diagnostics to identify potential mutational models can improve the evaluation of this neoplasm.

## 1. Introduction

Primary thyroid lymphoma (PTL) accounts for 1 to 5% of all thyroid malignancies and up to 2.5% of all extranodal non-Hodgkin lymphomas [1,2,3]. The most common type of PTL is diffuse large B-cell lymphoma (DLBCL) (50–80%) or extranodal marginal zone B-cell lymphomas (EMZBCLs) of the mucosa-associated lymphoid tissue type (MALT lymphoma) (20–30%) [4], the latter being associated with long-lasting autoimmune thyroiditis [5,6,7]. Other types are follicular lymphoma (FL) (12%) of all non-Hodgkin lymphomas (NHLs); extraosseous (extramedullary) plasmacytoma (3–5%) [8,9]; Hodgkin’s disease (7%); Burkitt’s lymphoma (4%); and T-cell PTL (<1%) [9]. The mean age of PTL patients is 67 years (in contrast with secondary thyroid lymphoma, which occurs at younger ages); the female-to-male ratio ranges from 2 to 8:1 [2,6,7]. Generally, PTL manifests as a rapidly enlarging neck mass with compressive symptoms and cervical lymphadenopathy [7]. The therapeutic approach should be conservative, without invasive procedures, being that the prognosis of these patients is basically poor [2,7,8].

## 2. Case Report

A 67-year-old man noticed a rapidly growing palpable neck mass corresponding to the right thyroid lobe. After the initial practitioner and surgeon examination, the surgical excision was indicated, and a fine-needle aspiration cytology (FNAC) and core needle biopsy (CNB) were planned, to better tailor the surgical approach (i.e., total thyroidectomy or not). In particular, for the FNA, a 23-gauge needle was utilized, while for the CNB, a 21-gauge cutting needle was employed, as previously described [10,11]. Two needle passes were made for both biopsies. Before CNB, a local anesthetic was administered. CNB has been reported as more accurate than FNAC in the diagnosis of thyroid nodules. In particular, the second-line use of CNB can diagnostically assess those thyroid lesions with previous inadequate or indeterminate cytology [12]. Previous meta-analyses of CNB evaluated its diagnostic accuracy based on its sensitivity or specificity for the diagnosis of malignancy or focused on specific conditions, such as nodules with non-diagnostic or atypia of undetermined significance/follicular lesion of undetermined significance (AUS/FLUS) results of previous FNA [12]. In general, thyroid CNB has been reported as a safe procedure, and the complication rate and comfort degree using CNB were not significantly higher than those recorded by FNAC [13]. CNB sampling allows for the obtaining of a tissue fragment with a size up to 500 µ and length up to 1.5 cm. These specimens should be the optimal material for extensive studies and ancillary techniques. In fact, on one hand, the micro-histologic examination detects nuclear changes, architectural alterations of follicular structures, and pathologic relations between adjacent tissues; on the other hand, the paraffin core sections permit automated immunostaining with high reproducibility [14]. The reproducibility and standardization of the reaction product are usually developed for immunohistochemistry (IHC) on histological sections, whereas dedicated recommendations and practice paradigms are still lacking for cytological samples (conventional smears, thin layer cytology, and cell blocks) [15].

Clinical data known before FNAC and CNB included: (1) autoimmune hypothyroidism on levothyroxine therapy (0.1 mg/d), with a current skewed TSH at 8.0 mlU/L; (2) growth of the lesion in 2 days to a solid mass; (3) onset of hoarseness and dyspnea; (4) at ultrasound (US), replacement of the right thyroid lobe by a 6 cm nodule, with poorly defined edges, strong hypoechogenicity, sparse internal echoes, heterogeneous and mild peripheral vascularization, and intense posterior echo (Figure 1); (5) at computed tomography (CT), an irregular mass in the right thyroid lobe, measuring 70 × 55 × 55 mm, displacing the trachea to the left and surrounding it dorsally in relation to the esophagus and caudally immersing itself in the anterior mediastinum, without recognizable cleavage plans (Figure 2A); (6) at contrast-enhanced magnetic resonance (MR), a structural inhomogeneity of the mass, some colliquate areas in the central portion, and post-contrast enhancement. Better defined was the extent of posterior involvement at the retropharyngeal and prevertebral space level, with the involvement of the posterior right portion of the larynx, of the thyroid cartilage, and of the cricoid, where a lack of adipose cleavage with the esophagus was evident (Figure 2B). 

Following the above information, at the time of FNAC, several diagnostic hypotheses could be considered: papillary thyroid carcinoma, poorly differentiated thyroid carcinoma, follicular thyroid carcinoma, medullary thyroid carcinoma, anaplastic thyroid carcinoma, goiter due to the uncorrected autoimmune hypothyroidism, metastasis from other organs, primary thyroid lymphoma, or lymphoma invading the thyroid gland. Both the cytopathologist and the surgeon were aware that the reliability of FNAC in these lesions was quite different, being high in papillary carcinoma and metastases from other cancers, and low/poor in the other conditions. Calcitonin was proven normal, and therefore, the scenario of a medullary carcinoma was virtually not possible. Then, FNAC was performed, and the samples showed a discrete population of round lymphocytes, two to three times larger than normal mature lymphocytes, and nuclei with finely dispersed chromatin, with scanty and pale cytoplasm. Occasional clusters of thyroid follicular cells were seen. The FNAC report was of the high-risk indeterminate category, namely TIR3B, according to the Italian classification [9]. The impressive and rapid clinical–instrumental changes led to a second FNAC together with a tru-cut biopsy. An FNAC CD45(+), oriented towards a monoclonal proliferation of lymphocyte cells, is shown (Figure 3A,B). Hematoxylin and Eosin (HE) staining of the core needle biopsy showed a population of atypical large centroblasts with round nuclei, vesicular chromatin, eosinophilic nucleoli, and basophilic cytoplasms (Figure 3C), intensely CD20(+) and CD3(−) (Figure 3D). p53(+) was shown in 100% of the neoplastic population, with a high growth fraction (Ki67/MIB 1: 80 < 90% of the cell population) (Figure 3E). Small atypical B-lymphocytes, monocytoid-like B-cells, were also detected closely to centroblastic proliferation. These cells showed aggression towards the thyroid follicular structures, as occurs in lymphoepithelial lesions. The diagnosis was diffuse large B-cell lymphoma (DLBCL), probably as large cell transformation of low-grade EMZBCL (Figure 3F,G). The patient was referred to oncology, and chemotherapy (CHOP) for DLBCL, as per our institute’s protocol, was planned for him. For aggressive DLBCL, a combination of CHOP and radiotherapy is the standard treatment. So, after evaluation by the oncology staff, the treatment was based on chemotherapy (CHOP) with an anthracycline-containing multidrug regimen (CHOP/CHOP like). He tolerated the chemotherapy well. After chemotherapy, there was a partial remission of the thyroid lesion, and after one year of follow-up, his disease was well controlled. Tumoral DNA was extracted from 10 µm FFPE sections, previously micro-dissected in order to obtain a tumor cell density >75% and subsequently quantified by a Qubit 4.0 fluorometer DNA HS Assay (Thermo Fisher Scientific, Waltham, MA, USA). For targeted NGS analysis, Myriapod NGS Cancer Panel DNA (Diatech Pharmacogenetics, Lesi, Italy), a commercial kit detecting predictive diagnostic SNV somatic mutations clinically relevant in 17 cancer-associated genes, including both oncogenes and tumor suppressor genes, was employed on the MiSeq^®^ platform (Illumina San Diego, CA, USA) using a coverage >500×. All results were analyzed using the Myriapod NGS Data Analysis Software (version 5.0.8) (Diatech Pharmacogenetics), and a minimum variant allele frequency (VAF) of 5% was applied for variant filtering. Each variant was investigated for its potential pathogenic role, using the publicly available databases Catalogue of Somatic Mutations in Cancer (COSMIC) “https://cancer.sanger.ac.uk/cosmic (accessed on 10 April 2023)”, ClinVar “https://www.ncbi.nlm.nih.gov/clinvar/ (accessed on 10 April 2023)”, and the bioinformatic software VarSome “https://varsome.com/ (accessed on 10 April 2023)”.

The results of the analysis indicated the presence of an insertion in exon 20 of ERBB2 gene c.2313_2324dupATACGTGATGGC (p.Ala775_GLy776insTyrValMetAla) with a VAF of 37.84% and classified as likely pathogenic according to the ACMG guidelines (ClinVar ID 13875; dbSNP rs397516975).

## 3. Discussion

Diagnosing PTL is still a challenge. There is no typical clinical presentation, there are no specific US features, and the cytological examination hides many pitfalls. Therefore, when we are facing a patient with a rapidly growing neck mass, we should consider several scenarios. Figure 4 illustrates schematically the actual flow of management of these patients, from the clinical presentation to the final assessment of PTL. PTL usually mimics poorly differentiated and anaplastic thyroid carcinoma [7,8,9,16,17,18]. Differentiating these two entities is very important, as the treatment for PTL (and for DLBCL in particular), usually implicates only chemotherapy, while surgical resection is necessary for anaplastic carcinoma. Large cell lymphomas are more often recognized as malignant tumors, while small cell lymphomas usually require immunophenotyping of the cell population to be correctly identified as malignant. In the present case, different factors, such as monolateral presentation, a rapidly growing firm mass, and male gender, suggested undifferentiated thyroid carcinoma (UTC). Through FNAC, we excluded anaplastic cancer on the basis of a purely morphological evaluation. The specimen was highly cellular, composed predominantly of medium-to-large-sized lymphocytes, with finely granular chromatin, in contrast to the compact nuclei of the mature lymphocytes. The nucleoli were either small and multiple in a marginal location (centroblasts), or large in a central position (immunoblasts). Having a specimen with all these characteristics, a diagnosis of thyroid lymphoma of the diffuse large B-cell type could have been possible. We also performed a preliminary attempt to establish an immunophenotype, subsequently confirmed by tissue biopsy. 

The accuracy of the cytopathologic diagnosis of primary malignant lymphoma of the thyroid depends on several factors, such as the adequacy of the specimen and proper cytopreparatory technique, as well as the interpreter’s familiarity with their cytopathologic patterns. Lymphoma cells are very fragile and dry quickly unless wet-fixed immediately for Papanicolaou stain. May Grunwald Giemsa (MG.G.) or Diff-Quik stains reduce artifacts, and as the smears are dried in the air, the cytoplasmic paleness of lymphocytes colored in blue is highly obvious. In our experience, this procedure improves diagnostic accuracy. Zhang et al., in a retrospective review of FNAC to diagnose primary thyroid lymphomas, showed that FNA sensitivity in diagnosing PTL was 72% when FNA reports “suspicious for” and “diagnostic of” PTL were considered together [4]. Ota et al. reported that enhanced posterior echoes and well-defined echographic borders are useful in distinguishing lymphomas from other lesions, and lymphomatous from non-lymphomatous tissue [19]. The sensitivity and specificity of FNAC in the diagnosis of PTL is highly increased by other techniques such as flow-cytometry (FCM), immunocytochemical/immunohistochemical studies, or using a molecular approach [7].

In this regard, our finding of a pathogenic mutation for the ERBB2 gene allows us to draw some considerations. Data from the cBioPortal for Cancer Genomics database (https://www.cbioportal.org/, accessed 10 April 2023) show that ERBB2 mutations are present in 6.38% of diffuse large B-cell lymphoma cases. However, the mutation spectrum does not include insertion variants at the level of exon 20 of the gene (p.Ala775_GLy776insTyrValMetAla), as in our case. ERBB2 exon 20 insertions are most frequently detected in NSCLC, urothelial, biliary tract, and breast cancers. These pathogenic variants occur in the α-C helix and in the loop following the α-C helix (from residues 770 to 783) that act as a regulatory element determining the activation state of ERBB2 [20]. With regard to NSCLC, as indicated by cBioPortal, ERBB2 mutations are present in 3.49% of cases, and 90% are represented by exon 20 insertion. Furthermore, these types of mutations tend to be mutually exclusive with other oncogenic driver mutations [20]. Insertion mutations in ERBB2 exon 20 result in significantly reduced sensitivity to current therapies with tyrosine kinase inhibitors and antibodies targeting this receptor, and therefore, several new therapeutic strategies are currently being developed [21]. 

In this regard, a newly published study revealed the effectiveness of trastuzumab deruxtecan, an antibody–drug combination, on a variety of HER2-mutated hard-to-treat advanced solid tumors, indicating its potential as an agnostic tumor treatment for these different types of cancer [16]. In fact, the "agnostic" approach to cancer therapy requires that the most effective cure is chosen based on the presence of a specific molecular mutation in cancer, regardless of the tissue or histology of the tumor [22,23,24,25], and could potentially also be applied to PTL. However, the history of agnostic treatments is still rather short, and approval for their use by specific international and national agencies requires appropriate clinical trials.

## 4. Conclusions

On the basis of these considerations, it is expected that in the future, the adoption of a mutational screening in NGS will support the diagnosis and also provide more information on the biology, prognostic factors, and potential targeted therapy of the disease.

The finding of ERRB2 mutations, while by itself cannot be considered a specific or pathognomonic finding, can open, in similar cases, perspectives for a personalized therapy and can offer insights on the prognosis of the disease.

Until the routine adoption of these recent significant technological innovations in the medical field, diagnosing PTL is still a relevant challenge in clinical practice. Cytopathologists addressing patients with potential PTL should be aware of its clinical, laboratory, and imaging information. Therefore, it is equally crucial that the Molecular Tumor Board, which was established to address the enormous gap between clinical knowledge and the promise of molecular diagnostics in cancer practice, serves as the focal point of this organizational paradigm [26]. A multidisciplinary approach is therefore necessary, and the surgeon should avoid removing the thyroid until PTL is excluded.

## Figures and Tables

**Figure 1 jpm-13-01203-f001:**
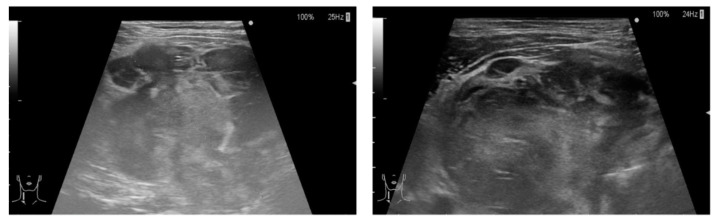
The preliminary ultrasound examination showing the presence of a hypoechoic nodule, with clear margins, of the right lobe of the thyroid, measuring 55 × 60 mm, without vascularity.

**Figure 2 jpm-13-01203-f002:**
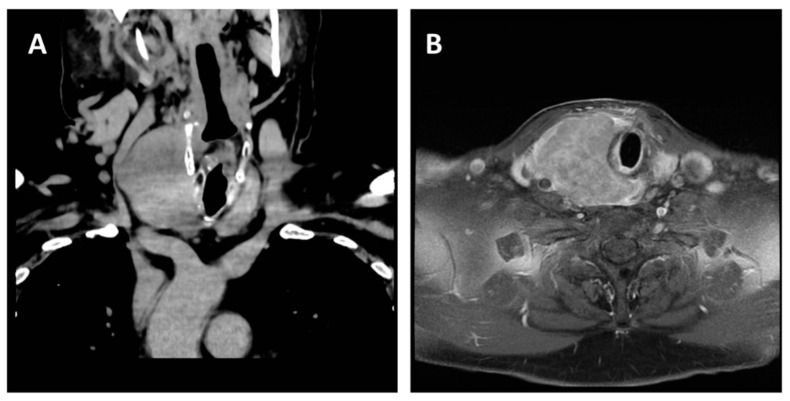
(**A**) CT showing an extensive tumor, 7 × 5 cm, of the right lobe of the thyroid, which displaces the trachea to the left and also surrounds it dorsally, relating to the esophagus and merging caudally into the anterior mediastinum. (**B**) MR showing solid tissue in correspondence with the right lobe of the thyroid (70 × 55 × 55 mm), no longer recognizable, extended caudally to the anterior-superior mediastinum within homogeneous signal intensity due to the presence of a fluid component in the context. This tissue causes compression and dislocation of the trachea, with a solid token at the level of the posterior wall, a possible site of infiltration, and extends cranially posteriorly at the level of the retropharyngeal and prevertebral space, with involvement of the right posterior portion of the larynx, of the thyroid cartilage, and of the cricoid.

**Figure 3 jpm-13-01203-f003:**
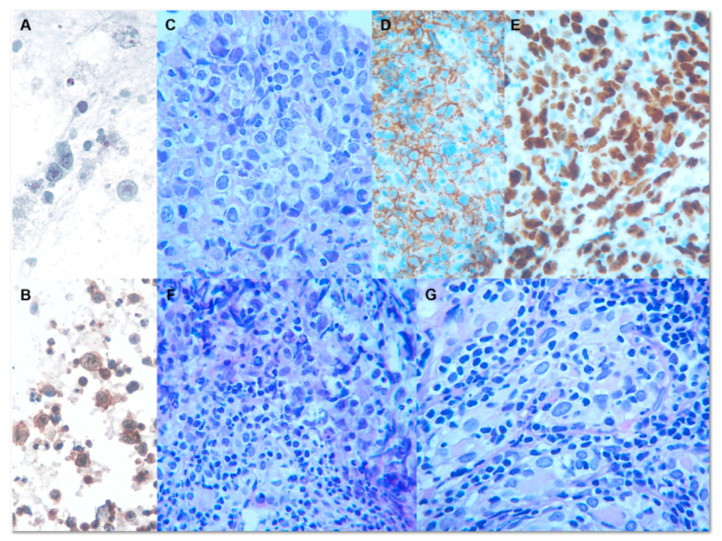
(**A**) Large, poorly differentiated lymphoid cells with large nuclei, open chromatin, and scanty cytoplasm (Papanicolaou staining 40×). (**B**) CD45 positivity oriented towards a lymphoid proliferation. (**C**) Diffuse large B-cell lymphoma component. (**D**) CD20 immunostain: large cells express CD20. (**E**) High proliferation index Ki67. (**F**) Transition zone: small atypical B-cells on left side and large cells of right side are closely admixed. The small B-cells reveal monocytoid-like features. (**G**) Small atypical monocytoid B-cells and lymphoepithelial lesion as seen in low-grade marginal zone B-cell lymphoma (extranodal MALT lymphoma of the thyroid gland).

**Figure 4 jpm-13-01203-f004:**
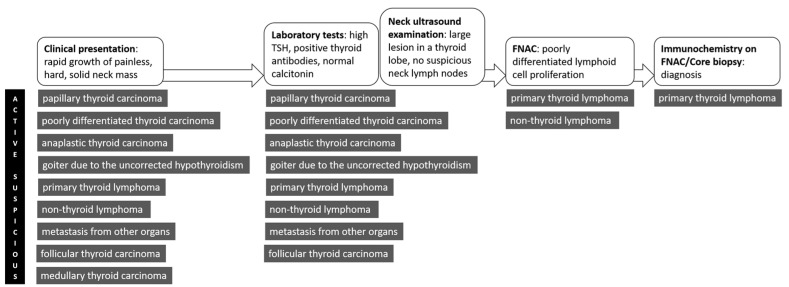
Theoretical timeline for the management of a patient with suspicious PTL.

## Data Availability

Data sharing is not applicable.

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
