# Peer review of "Primary Thyroid Lymphoma: How Molecular Biology and Ancillary Techniques Can Help the Cytopathologist Overcome This Diagnostic Challenge"

_jpm, 2023, doi:10.3390/jpm13081203_

Round 1
Reviewer 1 Report
This case report is an interesting account of how the final diagnosis of PTL was reached and the discovery of a mutation that was not typical for PTL. It contains general medical information and relevant symptoms of the patient. Symptoms, diagnosis, and some treatment are described, but the outcome requires further clarification to be more understandable for the readers.
Therefore, I have one minor concern:
After discovering the mutation, what were the subsequent treatment approaches and what was the outcome? The authors are discussing a personal approach to treating patients, but the legality of applying experimental treatments to patients makes it difficult (“agnostic” approach, pg 4 in the text: “the most effective cure is chosen based on the presence of a specific molecular mutation in cancer, regardless of the tissue or histology of the tumor”).
Author Response
We thank the reviewer for this useful suggestion. We apologize if we confused the reader by mentioning the topic of potential "agnostic approach" to pathology described in our case report. We only hypothesized a future "potential" treatment with agnostic therapy considering that the somatic mutation we found in the ERBB2 gene is under discussion in the recent international literature regarding a probable agnostic approach. In the text, we have clarified this point. In addition, according with the reviewers’ suggestions, we implemented the text by describing the treatment suggested by the current guidelines for diffuse large B-cell lymphoma (DLBCL) that the patient underwent and the outcome.
Reviewer 2 Report
This is a very interesting case report which provides diagnostic path for one of the most challenging problems in thyroid pathology.
I have a few suggestions to improve the paper:
1. Ultrasound, CT and MR scans should be included to demonstrate the patient's results.
2. As the authors underlined, the data on diagnostic path of such cases are scarce. However, the authors did not compare their case to most recently published results focused on usefulness of different methods including fine needle aspiration, core needle biopsy and open surgical biopsy. I believe that such comparison would be valuable.
Language editing is necessary.
Author Response
We thank the reviewer for this useful suggestion. According with the reviewers’ suggestions, Ultrasound, CT and MR scans figures have been added and the text was expanded by describing the topic suggested by the reviewer.